# Soil Moisture Retrieval Model for Remote Sensing Using Reflected Hyperspectral Information

**Jing Yuan [1,2], Xin Wang [1,2], Chang-xiang Yan [1,3,*], Shu-rong Wang [1], Xue-ping Ju [1,2] and Yi Li [1,2]**

[1]  Changchun Institute of Optics, Fine Mechanics and Physics, Chinese Academy of Sciences, Changchun 130033, China; 15543665143@163.com (J.Y.); percyet@126.com (X.W.); srwang@ciomp.ac.cn (S.-r.W.); juxueping14@mails.ucas.ac.cn (X.-p.J.); liyi_zju@163.com (Y.L.)

[2]  University of the Chinese Academy of Sciences, Beijing 100049, China

[3]  Center of Materials Science and Optoelectrics Engineering, University of Chinese Academy of Science, Beijing 100049, China

*  Correspondence: yancx0128@126.com; Tel.: +86-186-4307-5317

**Abstract:** The variation and the spatial–temporal distribution of soil water content have significant effects on heat balance, agricultural moisture, etc. A soil moisture (SM) retrieval model can provide a theoretical basis for realizing a rapid test and revealing the spatial–temporal variation of the surface water. However, remote sensors do not measure soil water content directly. Therefore, it is of great importance to establish a SM retrieval model. In this paper, the relationship between SM and diffuse reflectance was first derived using the absorption coefficient and scattering coefficient related to SM. Then, based on Kubelka–Munk (KM) theory, the SM retrieval model using reflectance information was further derived, which is a semi-empirical model with an unknown parameter obtained either from fitting or from experimental measurements. The validity and reliability of the model were confirmed with the validation set. The results showed that the root mean square errors of prediction (RMSEPs) of four soils were generally less than 0.017, while the coefficients of determination ($R^2$s) of four soils were generally more than 0.85, and the ratios of the performance to deviation (RPDs) of four soils were greater than 2.5 (470–2400 nm). Therefore, the model has high prediction accuracy, and can be well applied to the prediction of water content in different sorts of soils.

**Keywords:** hyperspectral remote sensing; soil moisture retrieval model; reflectance; semi-empirical model

## 1. Introduction

Soil moisture (SM) seriously affects the physical and chemical properties of soil and the growth of vegetation. The monitoring of SM plays a decisive role in crop yield estimation, drought monitoring, and evapotranspiration [1,2]. Compared with the traditional fixed-point monitoring method, the remote sensing technology can be used to realize the real-time and dynamic monitoring of SM in a large area. At present, there are four main methods of SM remote-sensing monitoring, namely thermal inertia [3,4], microwave remote sensing [5], the vegetation index [6], and hyperspectral remote sensing. By comparing these methods, the application of hyperspectral remote sensing in retrieving SM has attracted the attention of researchers [7–16], because it can be used to identify the absorption characteristics of SM at different spectral bands with high spectral resolution and spatial resolution.

In the light of the physical sense, three classes of optical models may be distinguished in this article, which are the semi-empirical model [17–21], physical model [22–27], and empirical/statistical model [28–32]. The key papers for these models are summarized in Table 1. The semi-empirical model has higher accuracy than the physical model, and it has more generality than the statistical/empirical

model. It not only uses measurement data, but also draws on basic physical principles [33]. Physical models are fully formulated based on the physics of radiative transfer. These models are usually exposed to input information that is difficult to determine. The empirical/statistical model is based on machine learning techniques such as support vector machine and artificial neural networks et al., and provides formidable tools for inferring the surface SM in complex/heterogeneous media. However, these models are in defect regarding the physical origin, and thus require a vast database for calibration. Consequently, studies on the quantitative retrieval of soil water content based on the semi-empirical model that have crucial significance are provided.

**Table 1.** Summary of three classes of optical models.

| Optical Model | Author | Key Results |
|---|---|---|
| Semi-empirical model | Liu et al. (2004) | Adopted the methods of relative reflectance, first-order differential and difference in the prediction and modeling of soil surface moisture. |
| | Deng et al. (2004) | Established a moisture content model of rough soil based on the physical process of soil spectral scattering. |
| | Whiting et al. (2004) | Fitted an inverted Gaussian function to the continuum and calibrated the area below the curve to soil moisture content (SMC). |
| | Yang et al. (2011) | Established the soil water parametric (SWAP)–Hapke model by introducing the soil water content information into the soil bidirectional reflectance model (SOILSPECT). |
| | Sadeghi et al. (2015) | Proposed a model based on the Kubelka–Munk (KM) two-flux radiative transfer model. |
| | This study | Established a SM retrieval model using the absorption coefficient and scattering coefficient related to SM based on KM theory. |
| Physical model | Ångström (1925) | Proposed a simple model where a wet soil is regarded as a dry soil covered with a thin film of liquid water. |
| | Lekner and Dorf (1988) | Improved the Ångström model by using the Fresnel coefficients instead of Snell's law |
| | Bach and Mauser (1994) | Introduced the Beer–Lambert–Bouguer law to account for light absorption in the water layer and extended the model to the visible light-short-wave infrared (VIS-SWIR) |
| | Kimmel and Baranoski (2007) | Published a ray tracing model called SPLITS (spectral light transport model for sand) |
| | Philpot (2010) | Proposed a simple waterborne soil reflectance spectrum simulation model |
| | Sun et al. (2015) | Improved the Philpot model by exploring the relationship between the soil water content and two parameters |
| Empirical/statistical model | Zaman et al. (2012) | Established a moisture content model using relevance vector machines and support vector machines |
| | Hassan Esfahani et al. (2015) | Developed an artificial neural network (ANN) model to quantify the effectiveness of using spectral images to estimate surface SM |
| | Wang et al. (2017) | Proposed a soil near-infrared spectroscopy prediction model based on deep sparse learning |
| | Wang et al. (2015) | Developed a regression model between polarization and SM. |
| | Wu et al. (2015) | Developed a SM prediction model with multiple linear regression, principal component regression, and partial least-squares regression, respectively |

However, the accuracy of the semi-empirical SM retrieval model remains to be raised. To solve this problem, we present a concise model with physically definable parameters based on the KM theory. The model is designed to describe the diffuse scattering of the absorbing and scattering medium. In previous studies, the diffuse reflectance in the Kubelka–Munk (KM) model is usually regarded as a parameter that needs to be inverted or constant for a given material and illumination wavelength. Nevertheless, it is found that diffuse reflectance is not only related to the material and wavelength, but also to the soil water content. The reason is that the absorption and scattering coefficients of soil are both affected by the soil water content, and diffuse reflectance is the function of the absorption and scattering coefficients based on the KM model. The model could estimate the water content of different sorts of soils with higher accuracy than before.

In this study, we aimed to: (1) investigate the influence of SM on the reflectance spectra; (2) present a concise model with physically definable parameters, which has good applicability and high accuracy, based on the KM theory; (3) verify whether the model can estimate the water content of different sorts of soils with high accuracy.

## 2. Materials and Methods

### 2.1. Soil Sample Preparation and Rewetting Experiment

In 2016, black soil samples were collected in Qiqihar (126°40′18.71″E, 47°37′18.28″N, Hei Long-jiang Province). In 2018, loessial soil, forest brown soil, and agricultural brown soil samples were collected in Changchun (respectively from 125°24′9.26″E, 43°47′8.27″N; 125°26′34.3″E, 43°47′6.2″N; and 125°25′36.86″E, 43°47′15.71″N, Ji Lin Province). The major types of land are forestland and farmland. For each kind of soil, the soil samples come from the same sampling site, so it is considered that the same kind of soil has the uniform soil particle property (i.e., mineral composition, organic matter, nutrients, etc.), ignoring the influence on the reflectance spectra of slight differences in organic matter, etc. The granulometric analysis with the mineralogical composition of four soils is shown in Table 2. The collected soil samples were further air-dried, and crushed to pass through a one-mm sieve so that stones, roots, and the vegetation litter were avoided from soils.

**Table 2.** Granulometric analysis with the mineralogical composition of four soils.

| Soil Name | Sand Contents/% | Silt Contents/% | Clay Contents/% | Mineralogical Composition |
|---|---|---|---|---|
| Black soil | 40.63 | 24.87 | 34.50 | Illite, montmorillonite |
| Forest brown soil | 19.22 | 40.06 | 40.72 | Hydromica, montmorillonite, and kaolinite |
| Agricultural brown soil | 26.92 | 45.01 | 28.07 | Hydromica, kaolinite, and vermiculite |
| Loessial soil | 28.36 | 17.89 | 53.75 | Kaolinite, illite, and montmorillonite |

The soil water content in this literature refers to the weight water content (the ratio of the weight of water in the soil to the weight of dry soil). Prior to the rewetting experiment, all of the soil samples were oven-dried at 105 °C for 24 h to eliminate soil water. Approximately 100 g of oven-dried soil for each sample was weighed using a scale (accuracy = 0.01 g) in the laboratory, and then placed in a petri dish. In order to prepare samples with different humidity gradients, they were wetted with different amounts of water. Water was sprayed into each soil sample while stirring so that the soil and water were fully mixed. After spraying, the soil sample was placed in a sealed bag with a good sealing effect and kept for 24 h, with the consequence that the soil could fully absorb water (Figure 1). The SM could be calculated from the amount of the water added. As a result, black soil samples, loessial soil samples, forest brown soil samples, and agricultural brown soil samples were prepared with 15 different SM levels, 14 different SM levels, 15 different SM levels, and 16 different SM levels, respectively.

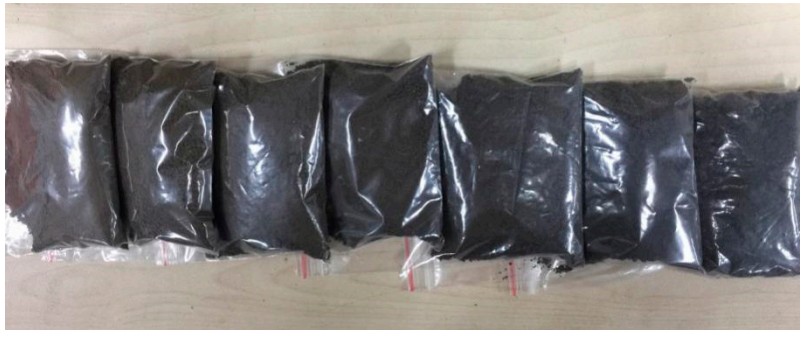

**Figure 1.** Rewetting soil samples.

## 2.2. Spectral Measurement and Pre-Processing

The hyperspectral reflectance data were acquired in a dark room using an ASD FieldSpec.3 Portable Spectrometer (Analytical Spectral Devices, Boulder, CO, USA). The main geometric parameters of the spectrometer set-up were illustrated as follows (Figure 2): a 50-W halogen lamp as the unique light source with a 30° incident angle was used to reduce the shadow effect caused by soil roughness.The lamp was set 10 cm away from the petri dish; the probe was mounted vertically about five cm above the dish, and the field angle of the probe was one degree. The soil depth was one cm. In order to obtain the absolute reflectance, the reflectance was standardized using a white Spectralon reference panel [34]. The arithmetic average of 10 spectral curves collected from each soil sample was regarded as the actual reflectance spectrum data.

Before the original spectral data were exported, splice corrections were calculated using view Spec™ software (version 6.0.0, ASD Inc., Longmont, CO, USA) to solve the breakpoint phenomena around 1000 nm and 1800 nm. The reflectance of each spectrum was narrowed to 470–2400 nm. To eliminate the noise in the spectra, the study applied RLOWESS smoothing to the original reflectance spectra curve.

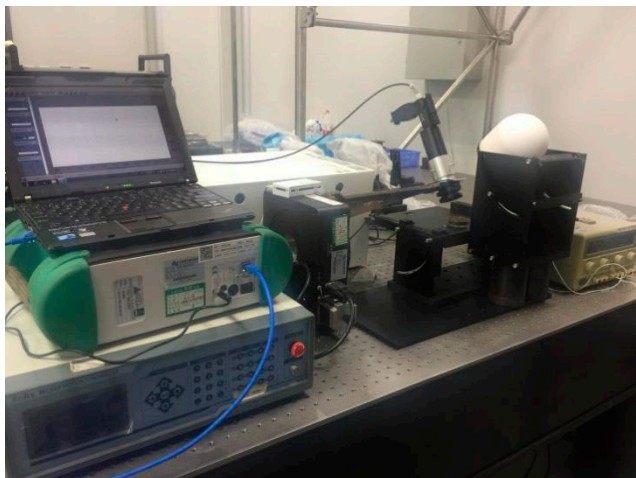

**Figure 2.** Experimental device.

## 2.3. KM Model

The KM [35] model describes radiative transfer, considering a downward and an upward light propagation flux (I and J, respectively), in an absorbing and scattering medium, perpendicular to the layer (Figure 3). The model assumes that (i) the layer exhibits an infinite lateral extension (so that the edge effects can be neglected); (ii) the light absorbing and scattering particles are uniformly distributed in the layer; (iii) particle dimensions are much smaller than the layer thickness, d, and (iv) the whole layer is homogeneously illuminated with a monochromatic diffuse light source [36,37].

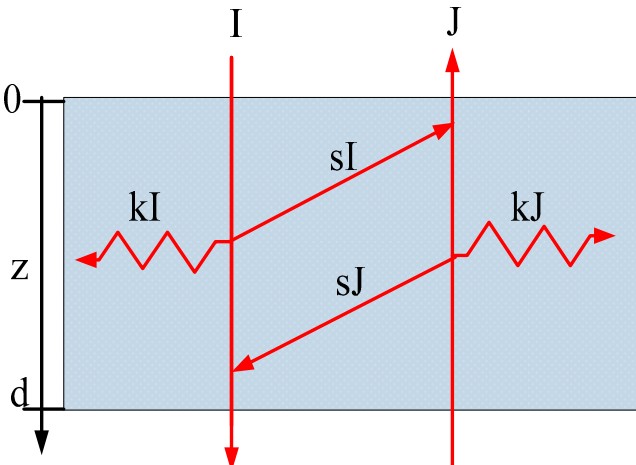

**Figure 3.** Visualization of the KM model. I and J are the two light fluxes in opposite directions; k and s are the absorption and scattering coefficients, respectively.

The KM model consists of two differential equations describing the light fluxes, $I(\lambda, z)$ and $J(\lambda, z)$, at a given wavelength, $\lambda$ (nm), and at a depth in the layer, $z$ (cm), with a light absorption coefficient, $k$ (cm$^{-1}$), and a light scattering coefficient, $s$ (cm$^{-1}$):

$$\frac{dI(\lambda, z)}{dz} = -(k+s)I(\lambda, z) + sJ(\lambda, z), \tag{1}$$

$$\frac{dJ(\lambda, z)}{dz} = (k+s)J(\lambda, z) - sI(\lambda, z). \tag{2}$$

By analytically solving these equations, reflectance ($R$) can be obtained [38]:

$$R = \frac{(1-\beta)^2[exp(\alpha d) - exp(-\alpha d)]}{(1+\beta)^2 exp(\alpha d) - (1-\beta)^2 exp(-\alpha d)}, \tag{3}$$

where $\alpha = \sqrt{k(k+2s)}$; $\beta = \sqrt{k/(k+2s)}$

With increasing layer thickness, d, the reflectance reaches the infinite reflectance value, $R_\infty$, which is used in diffuse reflectance spectroscopy, because a further increase of the sample thickness does not affect the measured signal. In this case, the calculation of the infinite reflectance in Equation (3) can be drastically simplified:

$$R_\infty = \frac{(1-\beta)}{(1+\beta)} = 1 + \frac{k}{s} - \sqrt{\left(\frac{k}{s}\right)^2 + 2\frac{k}{s}}. \tag{4}$$

By solving this equation for $k/s$, one gets the so-called Kubelka–Munk function:

$$r = \frac{k}{s} = \frac{(1-R_\infty)^2}{2R_\infty}. \tag{5}$$

*2.4. SM Retrieval Model*

For wet soil, reflectance, which is related to SM, mainly depends on diffuse scattering [39]. The relationship can be expressed as:

$$R = R_d = (1 - R_i)\frac{(1 - k_2)R_\infty}{1 - k_2 R_\infty}, \tag{6}$$

where $k_2$ is the Fresnel reflectance for diffuse light that exits the material and transits a thin layer–air interface at the material surface. In general, $k_2$ is a function of the surface roughness, refractive index,

and scattering angles. It is often assumed to be approximately equal to $R_i$ or treated as a constant [39]. $R_i$ is the Fresnel reflectance for light incident in air upon the target surface [21]:

$$R_i = \left( \frac{n_{water} - n_{air}}{n_{water} + n_{air}} \right)^2, \tag{7}$$

where $n_{water}$ is the refractive indices of water ($\approx 1.33$), and $n_{air}$ is the refractive indices of air ($\approx 1$).

Equation (6) can be rearranged as:

$$R_\infty(R) = \frac{R}{(1 - R_i)^2 + R \cdot R_i}. \tag{8}$$

Combining Equations (5) and (8) yields:

$$r(R) = \frac{k}{s} = \frac{(1 - R_\infty)^2}{2R_\infty} = \frac{\{1 - [\frac{R}{(1-R_i)^2 + R \cdot R_i}]^2\}}{\frac{2R}{(1-R_i)^2 + R \cdot R_i}}. \tag{9}$$

Variables $R_\infty$ and $r$ can be expressed mutually using Equations (4) and (5). The absorption and scattering coefficients are both affected by soil water content; thus, in the following modeling process, they will be equally considered as remotely sensible variables to estimate the SM.

Equation (9) shows that reflectance $R$ is affected by the absorption and scattering coefficients ($k$ and $s$) of the soil, because they are functions of the soil particle characteristics (i.e., mineral composition, organic matter, nutrients, etc.) and the soil water content. A frequently effective and commonly accepted assumption is that the absorption and scattering coefficients of a mixed medium can be regarded as a simple sum function of the absorption and scattering coefficients weighted by their composition proportions [38,40,41]. Given this assumption, we can describe the k and s of the soil surface as:

$$k(\theta) = k_{solid}(1 - \theta) + k_{water}\theta, \tag{10}$$

$$s(\theta) = s_{solid}(1 - \theta) + s_{water}\theta, \tag{11}$$

where $\theta$ is the soil water content, $k_{solid}$ and $k_{water}$ are the absorption coefficients of solid and water, and $s_{solid}$ and $s_{water}$ are the scattering coefficients of solid and water, respectively. The optical properties of soil water are different from pure water, as it contains not only pure water, but also dissolved organic matter and ions in addition to suspended particles, and the water itself is partially bound to the soil [26]. When the soil water content is $\theta_1$, the absorption and scattering coefficients of the soil, which are denoted as $k_1$ and $s_1$, can be written as:

$$k_1 = k_{solid}(1 - \theta_1) + k_{water}\theta_1, \tag{12}$$

$$s_1 = s_{solid}(1 - \theta_1) + s_{water}\theta_1. \tag{13}$$

Equations (10) and (11) can be written as:

$$k(\theta) = k_1 \left( \frac{1 - \theta}{1 - \theta_1} \right) + k_{water} \left( \frac{\theta - \theta_1}{1 - \theta_1} \right), \tag{14}$$

$$s(\theta) = s_1 \left( \frac{1 - \theta}{1 - \theta_1} \right) + s_{water} \left( \frac{\theta - \theta_1}{1 - \theta_1} \right), \tag{15}$$

Combining Equations (14), (15), and (5) yields:

$$r(\theta) = \frac{k(\theta)}{s(\theta)} = \frac{k_1\left(\frac{1-\theta}{1-\theta_1}\right) + k_{water}\left(\frac{\theta-\theta_1}{1-\theta_1}\right)}{s_1\left(\frac{1-\theta}{1-\theta_1}\right) + s_{water}\left(\frac{\theta-\theta_1}{1-\theta_1}\right)}. \tag{16}$$

The absorption and scattering coefficients of a soil sample, whether dry or wet, can be directly measured. Nonetheless, a more convenient and practical algorithm is one in which the numerator and denominator on the right side of Equation (16) are simultaneously divided by the scattering coefficient $s_1$:

$$r(\theta) = \frac{r_1\left(\frac{1-\theta}{1-\theta_1}\right) + a_1\left(\frac{\theta-\theta_1}{1-\theta_1}\right)}{\left(\frac{1-\theta}{1-\theta_1}\right) + a_2\left(\frac{\theta-\theta_1}{1-\theta_1}\right)}, \tag{17}$$

with:

$$a_1 = \frac{k_{water}}{s_1}, \tag{18}$$

$$a_2 = \frac{s_{water}}{s_1}, \tag{19}$$

$$r_1 = \frac{k_1}{s_1} = \frac{(1-R_1)^2}{2R_1}, \tag{20}$$

where $R_1$ is the reflectance of the soil in which the water content is $\theta_1$.

Mainly due to the strong absorption of water in the soil, the scattering of water in the soil is very weak, and even can be ignored compared with the scattering of water-bearing soil. Thus, $a_2 = 0$ [15], the model contains only one unknown parameter, and Equation (17) can be simplified to:

$$r(\theta) = \frac{r_1\left(\frac{1-\theta}{1-\theta_1}\right) + a_1\left(\frac{\theta-\theta_1}{1-\theta_1}\right)}{\left(\frac{1-\theta}{1-\theta_1}\right)}. \tag{21}$$

For remote-sensing applications, one needs to retrieve the soil water content from the reflectance data. For such an application, Equation (21) can be solved explicitly for soil water content as:

$$\theta(R) = \frac{\frac{r(R)-r_1}{a_1} + \theta_1}{\frac{r(R)-r_1}{a_1} + 1}, \tag{22}$$

with:

$$r_1 = \frac{k_1}{s_1} = \frac{(1-R_1)^2}{2R_1}, \tag{23}$$

$$r(R) = \frac{\left\{1 - \left[\frac{R}{(1-R_i)^2 + R \cdot R_i}\right]^2\right\}}{\frac{2R}{(1-R_i)^2 + R \cdot R_i}}. \tag{24}$$

### 2.5. Calibration and Validation

Data sets partitioning methods include the concentration gradient, random sampling, Kennard–Stone (KS), the sample set partitioning based on the joint x–y distance (SPXY), and the concentration gradient method that is used in this paper.

Different sorts of soil were similarly treated as follows: the whole set (n = 14, 15, or 16) was sorted in ascending order according to the SM level; one sample was selected as $\theta_1$, and we used a stratified sampling approach to separate the samples into four strata with three or four intervals, and one sample was selected from each stratum as an independent validation set. The remaining samples were selected as a calibration set.

The root mean square error of prediction (RMSEP), $R^2$, and ratio of the performance to deviation (RPD) between the predicted and measured SM were selected to evaluate the model performance. Generally, the larger $R^2$, the RPD, and the smaller RMSEP were indicators of a superior model. Interpretations of RPD values were classified into five classes: RPD < 1.4 indicated unacceptable models/predictions; $1.4 \leq$ RPD < 1.8 indicated fair models/predictions; $1.8 \leq$ RPD <2.0 indicated good models/predictions; $2.0 \leq$ RPD < 2.5 indicated very good models/predictions; and RPD $\geq$ 2.5 indicated excellent models/predictions [42,43]. All of the data analyses were carried out in Matlab R2014b (The Math Works Inc.: Natick, MA, USA).

### 2.6. Unknown Parameter Acquirement

The unknown parameter $a_1$ is the ratio of $k_{water}$, the absorption coefficient of soil water, to $s_1$, the scattering coefficient of soil with a water content of $\theta_1$. $a_1$ is wavelength dependent, because $k_{water}$ is dependent on the wavelength. It needed to be acquired according to the calibration set based on a least-squares algorithm. The best criterion for model parameter selection is to minimize the residual sum of squares between the simulated and the measured value. The optimization objective function is constructed as follows:

$$min\Delta R(\theta) = \sum (R_{measure} - R_{model})^2, \tag{25}$$

where $R_{measure}$ is the measured value for the laboratory, and $R_{model}$ is the theoretical value of the model. All of the data analyses were carried out in Matlab R2014b.

## 3. Results

### 3.1. Descriptive Statistics of SM

The summary statistics of SM for the whole, calibration, and validation sets are provided in Table 3, respectively, for four soils. For black soil, agricultural brown soil, and loessial soil, the values of the mean, standard deviation (SD), and coefficient of variation (CV) from three sets in every soil sample were relatively similar. As for forest brown soil, the calibration set varied from 0.08 (100 g of dry soil includes eight grams of water) to 0.22 with a CV of 30.76%, and the range in the validation set was from zero to 0.2 with a CV of 68.51%. Generally speaking, the characteristic statistics of both the calibration and the validation set were similar to the whole set, indicating that they were well divided to represent the whole set. In addition, the moisture contents of the various samples were relatively uniformly distributed across their range.

**Table 3.** Statistical description of SM contents for four sorts of soils. CV: coefficient of variation.

| Soil | Dataset Name | Number | Maximum | Minimum | Mean | SD | CV (%) |
|---|---|---|---|---|---|---|---|
| Black soil | whole | 14 | 0.24 | 0 | 0.1421 | 0.0662 | 46.6 |
| | calibration | 10 | 0.24 | 0 | 0.1460 | 0.0701 | 48.02 |
| | validation | 4 | 0.2 | 0.06 | 0.1325 | 0.0640 | 48.28 |
| Forest brown soil | whole | 14 | 0.22 | 0 | 0.1393 | 0.0566 | 40.66 |
| | calibration | 10 | 0.22 | 0.08 | 0.1430 | 0.0440 | 30.76 |
| | validation | 4 | 0.2 | 0 | 0.1300 | 0.0891 | 68.51 |
| Agricultural brown soil | whole | 15 | 0.23 | 0 | 0.1493 | 0.0649 | 43.43 |
| | calibration | 11 | 0.23 | 0 | 0.1418 | 0.0698 | 49.24 |
| | validation | 4 | 0.22 | 0.1 | 0.1700 | 0.0510 | 30.00 |
| Loessial soil | whole | 13 | 0.18 | 0.04 | 0.1092 | 0.0427 | 39.10 |
| | calibration | 9 | 0.18 | 0.04 | 0.1089 | 0.0459 | 42.20 |
| | validation | 4 | 0.15 | 0.07 | 0.1100 | 0.0408 | 37.11 |

### 3.2. Reflectance Spectral Feature of Four Soils at Different SM Levels

To analyze the influence of SM on the reflectance spectra, the spectral reflectance of four soils at different SM levels was partially investigated (Figure 4). The spectral curves at different SM levels showed similar shapes, which were parallel in substance, but with different intensities. Three obvious absorption peaks around 1420 nm, 1940 nm, and 2200 nm were exhibited in all of the SM levels. The difference of spectral reflectance for four soils at different SM levels depended on the location and depth of the shift of water absorption peak. The visible bands were less sensitive to changes in SM than the short-wave infrared (SWIR) bands. From Figure 4, it could be seen that the reflectance spectra of different sorts of soils were diverse, which was mainly because the soil particle characteristics (i.e., mineral composition, organic matter, nutrients, etc.) of different sorts of soils were distinct. The diversity of the soil optical properties exhibited in Figure 4 illustrate how the proposed models work for a range of soils.

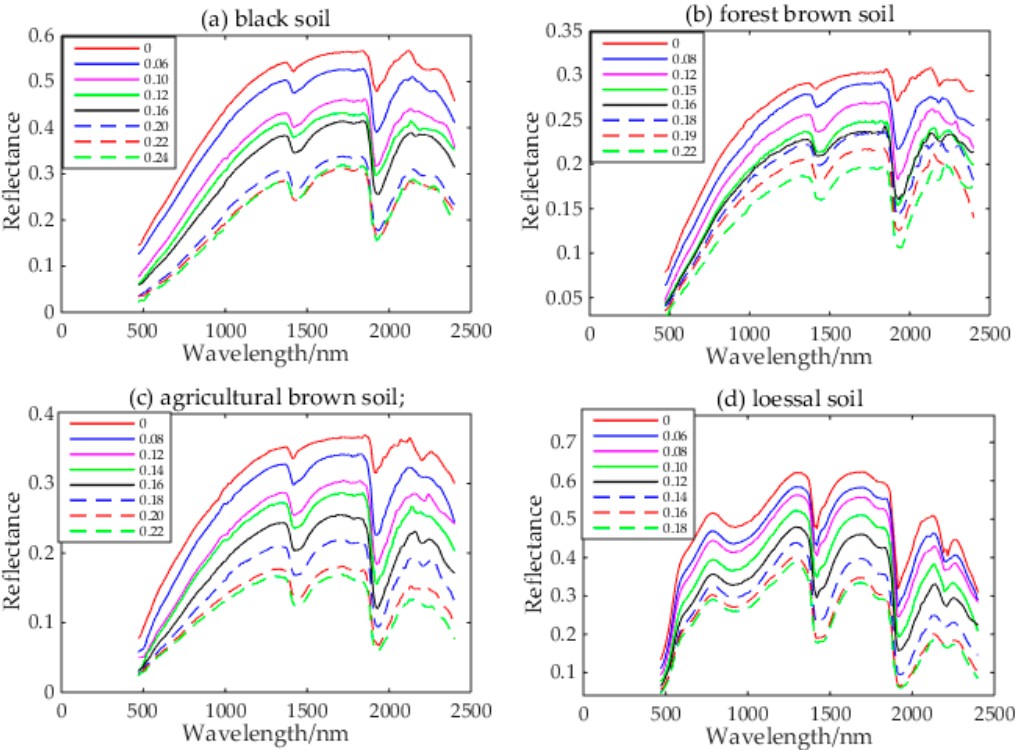

**Figure 4.** Reflectance spectra of four soils at different soil moisture (SM) levels: (**a**) black soil; (**b**) forest brown soil; (**c**) agricultural brown soil; and (**d**) loessial soil.

The visible bands were less sensitive to changes in SM than the SWIR bands. The study from Liu et al. [44] also pointed out that the change of soil spectral reflectance with water content in the SWIR band was significantly larger than that in the visible band, and the effect of water content on the soil spectral reflectance in the SWIR band was stronger than that in the visible band. The reflectance in the visible light band tends to saturate the change of soil water content. In the visible light band, the only role that water plays is to change the relative refractive index of the soil particle surface. With the increase of the soil water content, water is absorbed to the surface of the soil particle first, and then filled into smaller and larger pore sizes in turn. Therefore, once enough water is absorbed on most of the surface soil particles, the remaining water is filled into larger pores, which has little effect on the reflectivity of the visible light band. On the contrary, the SWIR spectral reflectance varies greatly with the increase of SM.

### 3.3. SM Retrieval Model

The $\theta_1$ values of black soil, forest brown soil, agricultural brown soil, and loessial soil were respectively 0.04, 0.04, 0.08, and 0.06. The unknown parameter $a_1$ was acquired by the least-squares algorithm combining the calibration set, wavelength-by-wavelength, in the range of 470 to 2400 nm (Figure 5).

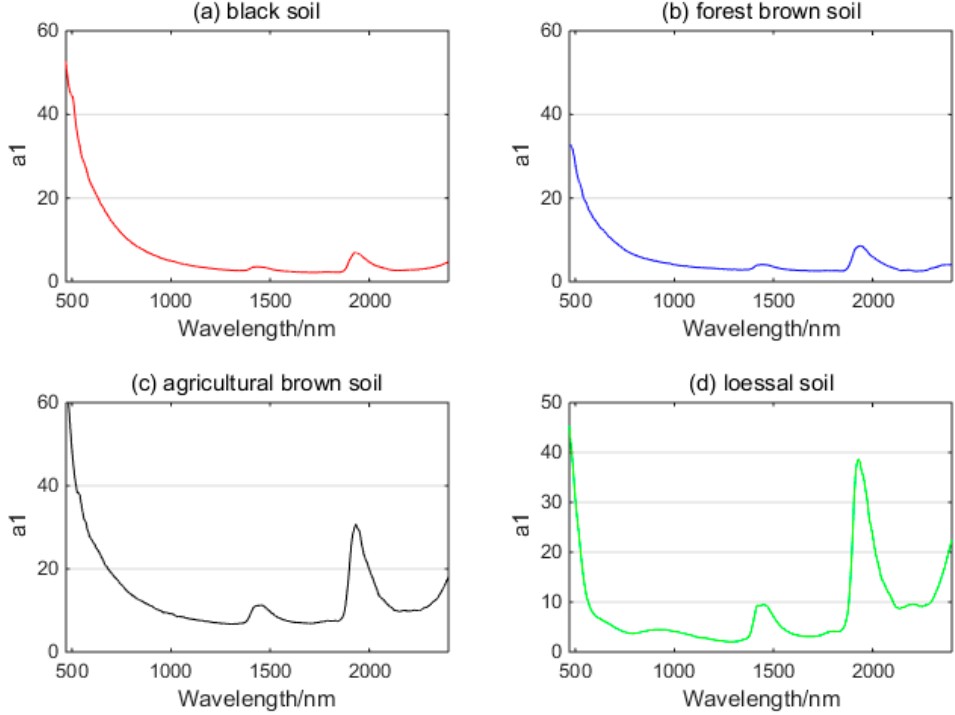

**Figure 5.** Parameter $a_1$ at different wavelengths for four soils: (**a**) black soil; (**b**) forest brown soil; (**c**) agricultural brown soil; and (**d**) loessial soil.

### 3.4. SM Estimation

Using the model mentioned in Equation (22), we could estimate the SM with a validation set. The retrieved result is displayed in Figure 6. The RMSEPs between the estimated and measured SM were computed wavelength-by-wavelength in the range of 470 to 2400 nm. Figure 6a is a spectrum diagram of the RMSEPs of black soil, which were generally less than 0.015, and the RMSEP at 1917 nm was 0.009879, which was the smallest. As for forest brown soil, the RMSEPs, which are shown in Figure 6b were generally less than 0.015, and the RMSEP at 1930 nm was 0.00665, which was the smallest. As shown in Figure 6c, the RMSEPs of agricultural brown soil were generally less than 0.017, and the RMSEP at 520 nm was 0.01145, which was the smallest. As for loessial soil, the RMSEPs, which are shown in Figure 6d, were generally less than 0.014, and the RMSEP at 608 nm was 0.007487, which was the smallest. In summary, the model had high prediction accuracy and could be well applied to the prediction of moisture content in different sorts of soils.

The $R^2$ values between the estimated and measured SM were computed in the range of 470 to 2400 nm, wavelength by wavelength. The $R^2$ values of black soil, forest brown soil, agricultural brown soil, and loessial soil were generally more than 0.925 (Figure 7a), 0.96 (Figure 7b), 0.85 (Figure 7c), and 0.85 (Figure 7d), respectively. Thus, the model had high stability.

From Figure 8, the RPDs of the four soils were greater than 2.5. The model had good prediction ability, and it could be well applied to the prediction of moisture content of different sorts of soils.

In the range of 470 to 2400 nm, the study used four validation samples to confirm the retrieved results, wavelength by wavelength (Figure 9). It could be observed that there was a significant linear relationship between the SM derived from the estimation and the actual measurement results.

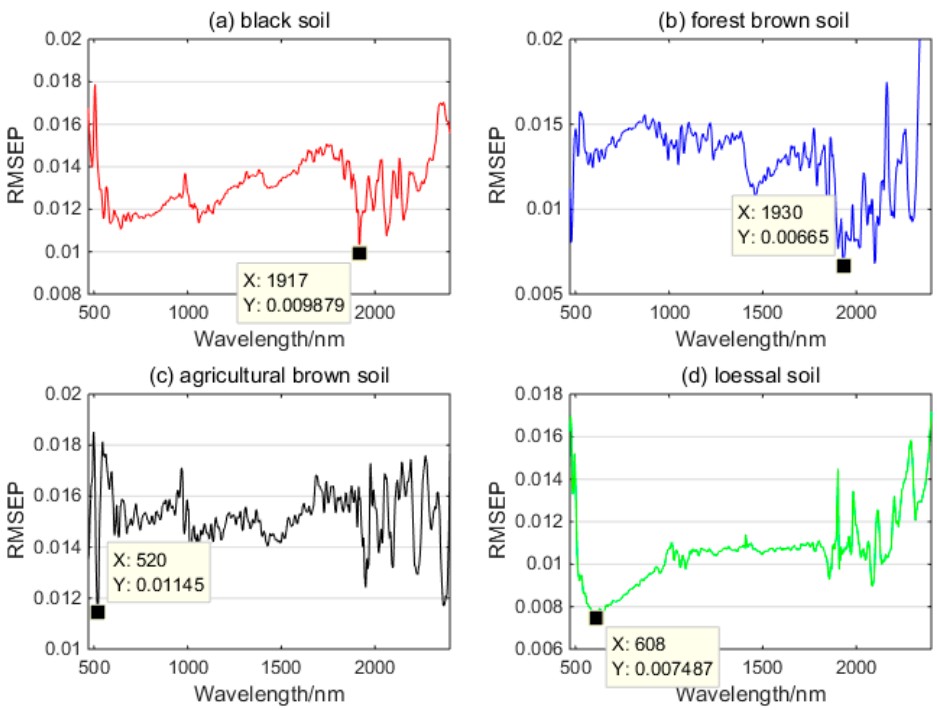

**Figure 6.** Root mean square errors of prediction (RMSEPs) at different wavelengths for four soils: (**a**) black soil; (**b**) forest brown soil; (**c**) agricultural brown soil; and (**d**) loessial soil.

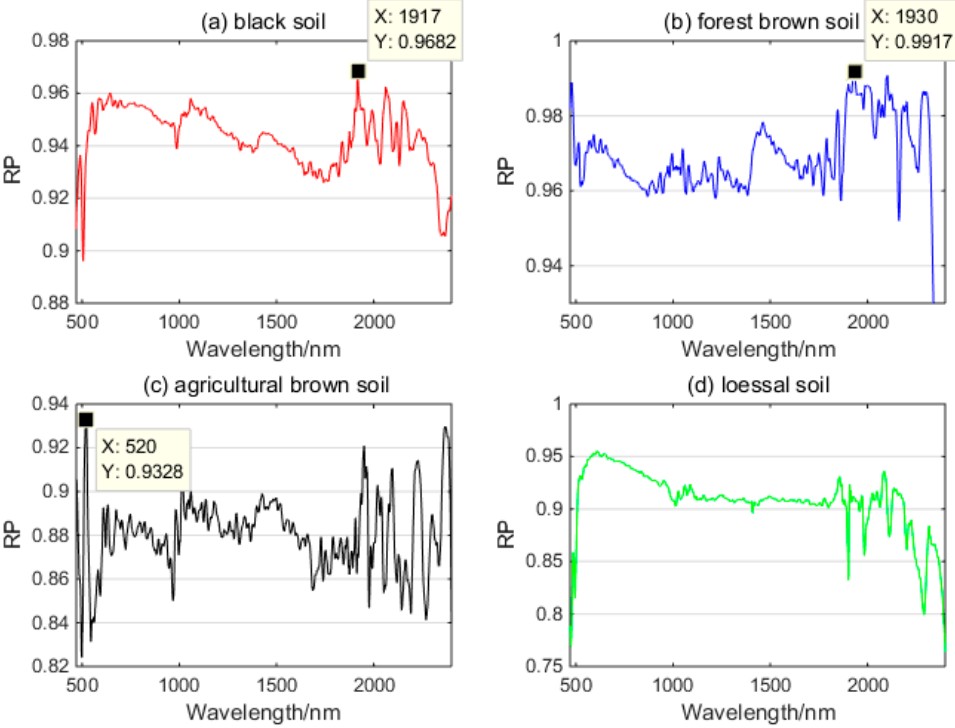

**Figure 7.** $R^2$ values at different wavelengths for four soils: (**a**) black soil; (**b**) forest brown soil; (**c**) agricultural brown soil; and (**d**) loessial soil.

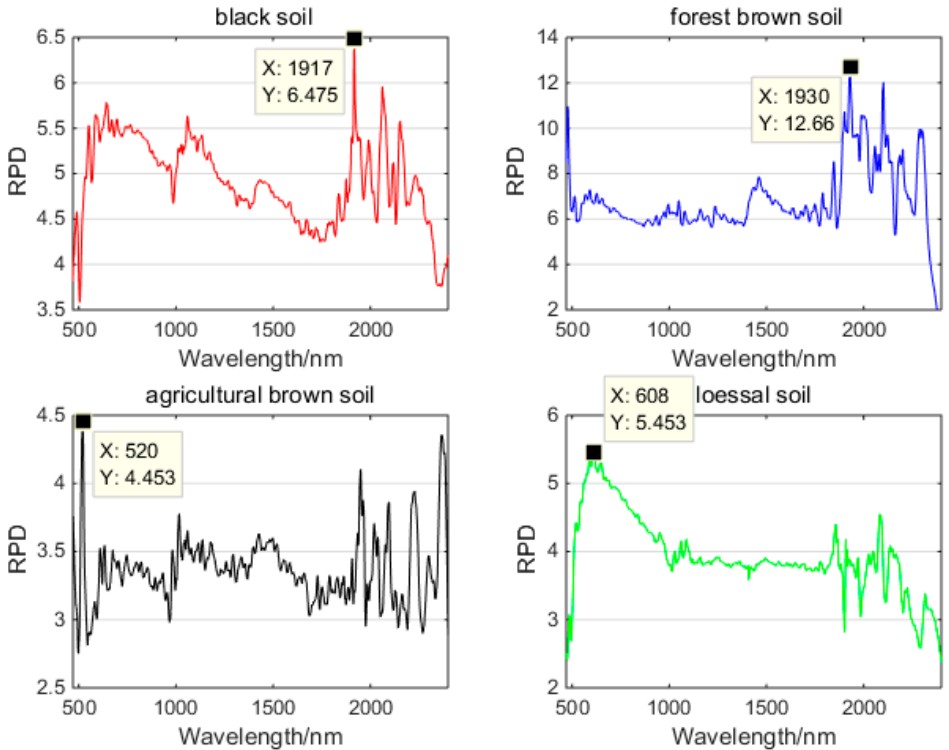

**Figure 8.** Ratios of the performance to deviation (RPDs) at different wavelengths for four soils.

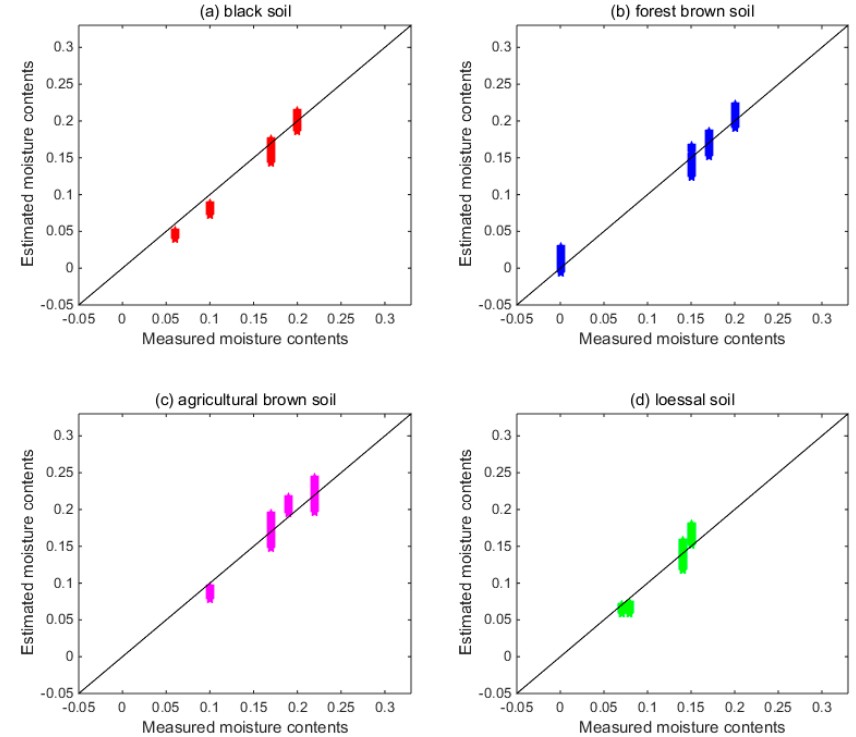

**Figure 9.** Comparison between SM from the retrieval model and the measurements for (**a**) black soil; (**b**) forest brown soil; (**c**) agricultural brown soil; and (**d**) loessial soil.

In order to concretely depict the accuracy of the model mentioned in Equation (22), the RMSEs of the measured and simulated moisture content of four validated samples of four soils were computed

in the range of 470 to 2400 nm (Table 4), which ranged from 0.0053 to 0.0218, it further showed that the model could estimate the moisture content of different soils with high accuracy.

**Table 4.** RMSE of four validation samples of four soils in the range of 470 to 2400 nm. RMSE: root mean square error.

| Soil | Moisture Contents | RMSE |
|---|---|---|
| Black soil | 0.06 | 0.0159 |
| | 0.10 | 0.0181 |
| | 0.17 | 0.0088 |
| | 0.02 | 0.0063 |
| Forest brown soil | 0.00 | 0.0218 |
| | 0.15 | 0.0130 |
| | 0.17 | 0.0054 |
| | 0.20 | 0.0062 |
| Agricultural brown soil | 0.10 | 0.0171 |
| | 0.17 | 0.0118 |
| | 0.19 | 0.0189 |
| | 0.22 | 0.0118 |
| Loessial soil | 0.07 | 0.0061 |
| | 0.08 | 0.0160 |
| | 0.14 | 0.0053 |
| | 0.15 | 0.0118 |

To further verify the validity of the model, four samples were randomly selected from the whole sets of the four soils as the validation sets, and the remaining samples were calibration sets. The unknown parameter $a_1$ in Equation (22) was inverted using the validation set, and then the SM was estimated by randomly selecting four verification samples. The RMSEPs between the estimated and measured SM were computed at six wavelengths corresponding to various bands of Landsat TM and ETM+ satellite, including band one (blue, 480 nm), band two (green, 560 nm), band three (red, 660 nm), band four (near infrared, 830 nm), band five (SWIR, 1650 nm), and band seven (SWIR, 2210 nm). The above procedure was repeated 50 times, and the minimum value, the maximum value, and the average value of RMSEPs were calculated at six wavelengths, respectively. The results (Table 5) show that the mean value of the RMSEPs of four soils ranged from 0.0084 to 0.0297, which reveals that the model can effectively invert the soil water content, and the partitioning of the validation set and the calibration set has an impact on the accuracy of the model, even though the impact is not great. Compared with the random sampling, the concentration gradient method for the data sets' partition can obtain relatively high-precision results.

**Table 5.** Statistical description of RMSEPs; these are calculated 50 times.

| Wavelength | | 480 nm | 560 nm | 660 nm | 830 nm | 1650 nm | 2210 nm |
|---|---|---|---|---|---|---|---|
| Black soil | Concentration gradient | 0.0148 | 0.0116 | 0.0119 | 0.0112 | 0.0147 | 0.0134 |
| | Minimum | 0.0106 | 0.0083 | 0.0087 | 0.0062 | 0.0107 | 0.0102 |
| | Maximum | 0.0335 | 0.0225 | 0.0220 | 0.0209 | 0.0254 | 0.0230 |
| | Mean | 0.0260 | 0.0149 | 0.0149 | 0.0152 | 0.0180 | 0.0182 |
| Forest brown soil | Concentration gradient | 0.0082 | 0.0135 | 0.0131 | 0.0150 | 0.0136 | 0.0117 |
| | Minimum | 0.0025 | 0.0059 | 0.0046 | 0.0069 | 0.0109 | 0.0099 |
| | Maximum | 0.0283 | 0.0188 | 0.0183 | 0.0201 | 0.0261 | 0.0358 |
| | Mean | 0.0168 | 0.0121 | 0.0117 | 0.0135 | 0.0220 | 0.0297 |
| Agricultural brown Soil | Concentration gradient | 0.0155 | 0.0176 | 0.0159 | 0.0154 | 0.0156 | 0.0135 |
| | Minimum | 0.0034 | 0.0067 | 0.0078 | 0.0054 | 0.0072 | 0.0121 |
| | Maximum | 0.0284 | 0.0236 | 0.0220 | 0.0215 | 0.0195 | 0.0275 |
| | Mean | 0.0141 | 0.0170 | 0.0164 | 0.0152 | 0.0161 | 0.0181 |
| Loessial soil | Concentration gradient | 0.0152 | 0.0087 | 0.0077 | 0.0090 | 0.0106 | 0.0121 |
| | Minimum | 0.0055 | 0.0048 | 0.0029 | 0.0040 | 0.0053 | 0.0075 |
| | Maximum | 0.0257 | 0.0151 | 0.0132 | 0.0125 | 0.0157 | 0.0144 |
| | Mean | 0.0160 | 0.0101 | 0.0086 | 0.0084 | 0.0121 | 0.0123 |

## 4. Discussion

KM theory is widely used to model the diffuse scattering behavior in the bulk of the material. Le Hors et al. [39] simplified the original KM theory by developing a diffuse scattering coefficient, which is the ratio of the diffuse scatter intensity to the scatter from a perfect Lambertian surface. Yang et al. [45] regarded the diffuse reflectance in the KM model as a parameter that needs to be inverted. Zhan et al. [46] regarded the diffuse reflectance in the KM model as a constant for a given material and illumination wavelength. However, the study found that diffuse reflectance was not only related to material and wavelength, but also to soil water content. Combining the absorption coefficient and scattering coefficient related to soil water content, the relationship between soil water content and diffuse reflectance was derived. Furthermore, the SM retrieval model using reflectance information was established, which is a semi-empirical model with an unknown parameter obtained either from fitting or from experiment measurements.

The unknown parameter $a_1$ was acquired by the least-squares algorithm combining the calibration set, wavelength by wavelength, in the range of 470 to 2400 nm (Figure 5). The trend of spectra was consistent with the SM absorption spectrum that was studied by William Philpot [26]. There are three moisture absorption bands which are around at 1420 nm, 1940 nm, and 2800 nm [26]. The spectrum of about before 1200nm was caused by the absorption of dissolved organic material, which was described by a model suggested by Bricaud et al. Absorption by the soil pore water is not the same as absorption by pure water [47,48]. The absorption spectrum of moisture was modified, which was probably due to absorption by substances dissolved in the moisture and/or as a result of water being partially bound to the soil [26].

The results of moisture content estimation indicated that the model could be well applied to the prediction of moisture content in different sorts of soils with high prediction accuracy (RMSEPs of four soils ≤0.017), high stability ($R^2$ values of four soils ≥0.85), and good prediction ability (RPDs of four soils ≥2.5). Figures 6–8 indicate a greater potential of SWIR bands than the visible and near-infrared bands for the remote sensing of SM, because reflectance values in the SWIR bands correspond to unique levels of SM. This study further verifies previous findings [19,49] that the SWIR wavelengths provide the optimal bands in the solar domain (i.e. wavelength between 350–2500 nm) for the remote sensing of SM. Compared with other soils, the prediction accuracy of loess is higher. The different behavior of the loessial soil may be related to its porosity and organic carbon content, which are significantly lower than those of the three other soils. The results in Table 5 show that for semi-empirical models, the partitioning of the validation set and the calibration set has an impact on the accuracy of the model, which is consistent with many previous studies. In order to obtain high-precision results, the partitioning of set algorithms such as the concentration gradient, KS, and SPXY are useful.

One significant advantage of the proposed model is that it has high prediction accuracy and can be well applied to the estimation of moisture content of different sorts of soils. Another crucial advantage of the model is that it is not restricted to a single band or wavelength. Properly calibrated, it is competent to describe the response of reflectance to water content over the full optical range. A distinct constraint of the model is that it contains an unknown parameter, and thus requires soil information a priori to be solved (i.e. calibration). This is not an unvanquishable prerequisite, given that our master intent in modeling the SM–reflectance relationship has been to remotely retrieve surface SM for environmental and agricultural applications. All of the optical methods will suffer from a shallow penetration depth of the optical bands, as well as limited applicability in vegetated soils.

## 5. Conclusions

In this literature, firstly, based on the KM theory, the relationship between spectral reflectance and diffuse reflectance was established, and then the relationship between soil water content and diffuse reflectance was derived using the absorption coefficient and scattering coefficient related to soil water content. Then, finally, an SM retrieval model using reflectance information was established, which is a semi-empirical model with an unknown parameter obtained either from fitting or from experiment

measurements. Combined with the spectral reflectance data measured by experiment, the model parameter was acquired based on the least square algorithm. The validity and reliability of the model were verified with an independent validation set. The results showed the following. (1) The RMSEPs of four soils between estimated and measured soil water content were generally less than 0.017, and in the range of 470 to 2400 nm. Comparing the accuracy of four soils, the highest accuracy was found in forest brown soil; its RMSEP at 1930 nm was 0.00665, which was the smallest. The $R^2$ values of four soils were generally more than 0.85. The RPDs of four soils were greater than 2.5. (2) The RMSEs of four validated samples of four soils that were computed in the range of 470 to 2400 nm ranged from 0.0053 to 0.0218. Therefore, the model has high prediction accuracy and can well be applied to the prediction of water content in different sorts of soils.

This study was the first step toward focusing on the theoretical aspects of the model and its testing under well-controlled laboratory conditions. Future studies are underway to examine/extend the model for field and large-scale applications when facing challenges such as high degrees of heterogeneity, surface roughness, topographical features, shadow effects, etc.

**Author Contributions:** Conceptualization, J.Y.; methodology, J.Y.; software, J.Y.; validation, J.Y., X.W., C.-x.Y.; formal analysis, J.Y.; investigation, J.Y.; resources, J.Y. and X.W.; data curation, J.Y.; writing—original draft preparation, J.Y.; writing—review and editing, X.W., C.-x.Y., S.-r.W., X.-p.J. and Y.L.; visualization, J.Y. and X.W.; supervision, C.-x.Y. and S.-r.W.; project administration, C.-x.Y.; funding acquisition, C.-x.Y.

**Funding:** This research was financially supported by the National Key Research and Development Program of China (2016YFF0103603) and National Natural Science Foundation of China (NSFC) (61627819, 61727818 and 6187030909).

**Conflicts of Interest:** The authors declare no conflict of interest.

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
