# Peer review of "Soil Moisture Retrieval Model for Remote Sensing Using Reflected Hyperspectral Information"

_remotesensing, doi:10.3390/rs11030366_

Round 1

Reviewer 1 Report

The manuscript describes a method for estimating soil moisture in bare soil samples by using thee Kubelka-Munk absorption \scattering theory to predict the reflectance spectra under various soil moisture condition, then comparing (validating) these predictions against hyperspectral measurements made under laboratory conditions. In general, I though the study was well designed (although with small sample sizes, see notes below) and relevant to the journal.  I have a few comments for the authors to consider.

The data set is small, especially the validation part of the sample (n=4).  Given the relatively small overall number of samples, I wonder if a bootrapping or leave-one-out approach might not be a better way to establish the accuracy and reliability of the method?  Cross-validation by multiple folding of the data set might also be used.  For example, the analysis could be repeated several times, each time holding out a random sample of four, then all runs compared to see if the validity holds up. I am especially concerned with using RMSE, R^2 , and RPD which are all parametric measures, and that the error bars for a single small sample may just be so large as to render the analysis meaningless.

While the theoretical background (especially the description of the K-M theory) were very well done, some of the methods were not entirely clear.  It's not all that clear how a1 was estimated, nor is it totally clear what it even is.  Is it wavelength dependent?  If it's estimated by least squares, then is the fit based on n=4, or is the least-squares fit derived from some other portion of the sample?  A little more clarity on this would help readers (like myself) who may eventually want to use this method actually be able to operationalize it.

Also in the methods, it is not entirely clear what the unit of soil moisture is. Context suggests that its proportional, or dimensionless. Is this correct?  Also, it would be helpful to know not just the range of values, but also their actual distribution within the test samples. Were the moisture contents of the various samples relatively uniformly distributed across their range, or were they uneven?  It would be useful to know.  

The study is done in the lab, under artificial conditions.  It would be useful for the authors to discuss how this method might be used operationally, and under what conditions it is likely to be valid.

In addition to these specific questions, I would suggest that the paper be thoroughly edited for correct English usage.  There are numerous examples of awkward usage throughout the manuscript.  A couple of especially obvious examples can be found in lines 144-146 and lines 15-16, but there are others and they should be corrected prior to publication.  

Author Response

Thank you for your pertinent comments on my article. I have provided a point-by-point response to these comments.

Reviewer 2 Report

I enjoyed reading the paper by Yuan et al, which attempted to estimate soil moisture using reflected hyperspectral information. It is of interest for the soil moisture community and I recommend it for publication after considering the following:. 

To help place the findings into a broader international context (in the Introduction section) can the authors provide an objective-focused tabular literature review of relevant research: including the current study as the last row in this proposed table. Doing this should make it more apparent to readers (and reviewers) what the new contribution of this study.  

For examples of objective-focused tabular literature reviews, please see Table 1 of the following papers.

Jarihani, A.A., et al (2015) Where does all the water go? Partitioning water transmission losses in a data-sparse, multi-channel and low-gradient dryland river system using modelling and remote sensing. Journal of Hydrology. 529(3), 1511-1529, doi:10.1016/j.jhydrol.2015.08.030

l40  backscatter and brightness temperature (kerr et al 2016, wagner et al 1999, Al yaari et al 2014.........etc)

L84 we hope ......   remove this

add a some sentences to describe the following sections

l87 what is the soil depth?

add some photos to support sections 2.1 and 2.2

There are lots of typos and the English must be improved.

Something is wrong with the order of the sentences 246-255

It was not clear to this reviewer how the authors seprated the samples for calibration and validation. I would add two sections one for calibration and the other for validation. 

Is this model transferable? 

Author Response

Thank you for your pertinent comments on my article.I have provided a point-by-point response to these comments.

Reviewer 3 Report

This paper is about the use of Kubelka-Munk theory to retrieve soil moisture from hyperspectral reflectance.

I suggest to send the paper to a lower impact journal due to the obviously of the analysis and related discussions.

The paper is not inserted very well in remote sensing context, so I suggest you to improve the references in the introduction related to the soil moisture retrieval algorithms mostly used in the others remote sensing branches. (row 40-42).

Furthermore I suggest you to better define the granulometric analysis with an adequate mineralogic composition of the samples introduced in this paper. (around row 90).

Please provide an application case study in which your retrieval of soil moisture should be very crucial.

Author Response

(The authors gave the same response as above.)

Round 2

Reviewer 3 Report

The manuscript is now ready to be published.